# Effect of Prolong-life-with-nine-turn-method (Yan Nian Jiu Zhuan) Qigong on fatigue and gastrointestinal function in patients with chronic fatigue syndrome: Study protocol for a randomized controlled trial

Yuanjia Gu[1,2☯], Yanli You[3☯], Guangxin Guo[1,2], Fangfang Xie[1‡], Chong Guan[1‡], Chaoqun Xie[1], Yanbin Cheng[4], Qing Ji[2], Fei Yao[1,2]*

1 School of Acupuncture-Moxibustion and Tuina, Shanghai University of Traditional Chinese Medicine, Shanghai, China, 2 Shanghai Municipal Hospital of Traditional Chinese Medicine, Shanghai University of Traditional Chinese Medicine, Shanghai, China, 3 Department of Traditional Chinese Medicine, Naval Medical University, Shanghai, China, 4 YueYang Hospital of Integrated Traditional Chinese and Western Medicine, Shanghai University of Traditional Chinese Medicine, Shanghai, China

☯ These authors contributed equally to this work.
‡ FF and CG also contributed equally to this work.
* doctoryaofei@126.com

**Data Availability Statement:** No pilot datasets were generated or analysed during the study

## Abstract

### Introduction

Chronic fatigue syndrome (CFS) is a physical and mental disorder in which long-term fatigue is the main symptom. CFS patients are often accompanied by functional gastrointestinal diseases (FGIDs), which lead to decreased quality of life and increased fatigue. Prolong-life-with-nine-turn-method (PLWNT) is a kind of physical and mental exercise. Its operation includes adjusting the mind, breathing and cooperating with eight self-rubbing methods and one upper body rocking method. PLWNT was used to improve the digestive function in ancient China and to treat FGIDs such as functional dyspepsia and irritable bowel syndrome in modern times. Previous studies have shown that PLWNT can reduce fatigue in patients with CFS. But it is unclear whether the effect of PLWNT on CFS fatigue is related to gastrointestinal function. The aim of this study was to explore the relationship between PLWNT and fatigue and gastrointestinal function in patients with CFS.

### Methods

This study is a non-inferiority randomized controlled trial (RCT). The whole study period is 38 weeks, including 2 weeks of baseline evaluation, 12 weeks of intervention and 6 months of follow-up. Ninety-six CFS patients will be stratified random assigned to the intervention group (PLWNT) and the control group (cognitive behavior treatment) in the ratio of 1:1 through the random number table generated by SPSS. In the evaluation of results, Multidimensional Fatigue Inventory-20 (MFI-20), Gastrointestinal Symptom Rating Scale (GSRS), Bristol Stool Form Scale (BSFS), and Short Form 36 item health survey (SF-36) will be

protocol. At the end of this study, the relevant research results will be published in the journal. The trial protocol, subject data and statistical analysis can be obtained by viewing journal articles, entering the clinical trial center or consulting relevant authors.

**Funding:** This trial is funded by Youth Qihuang Scholars Support Program, Future Plan for Traditional Chinese Medicine Inheritance and Development of Shanghai Municipal Hospital of Traditional Chinese Medicine (No.WLJH2021ZY-ZYY011; No.WLJH2021ZY-GZS007). Funders have no role in the design of this study and no role in its execution, analysis or interpretation of the data, or the decision to submit results. This work was supported by YueYang Hospital of Integrated Traditional Chinese and Western Medicine, Shanghai University of Traditional Chinese Medicine, Shanghai Municipal Hospital of Traditional Chinese Medicine.

**Competing interests:** The authors declare that they have no competing interest and the research was conducted in the absence of any commercial or financial relationships that could be construed as a potential conflict of interest.

evaluated at week 0 (baseline), week 6 (midterm), week 12 (endpoint) and month 9 (follow up). The intestinal flora will be evaluated at week 0 (baseline) and week 12 (endpoint). The data results will be processed by statistical experts. The data analysis will be based on the intention to treat principle and per-protocol analysis. In the efficacy evaluation, repeated measurement analysis of variance will be used for data conforming to normal distribution or approximate normal distribution. The data which do not conform to the analysis of repeated measurement variance will be analyzed by the generalized estimation equation Linear discriminant analysis will be used to clarify the difference species of intestinal flora. The significance level sets as 5%. The safety of interventions will be evaluated after each treatment session.

## Discussion

This trial will provide evidence to PLWNT exerting positive effects on fatigue and gastrointestinal function of CFS. It will further explore whether the improvement of PLWNT on CFS fatigue is related to gastrointestinal function.

## Trial registration

The trial was registered at Chinese Clinical Trial Registry http://www.chictr.org.cn/showproj. aspx?proj=151456 (Registration No.: ChiCTR2200056530). Date: 2022-02-07.

## Introduction

Chronic fatigue syndrome (CFS) is a physical and mental disease characterized by long-term or repeated fatigue. It includes a series of heterogeneous symptoms such as musculoskeletal pain, cognitive impairment, sleep disorder, gastrointestinal symptoms and so on [1]. Functional gastrointestinal disorders (FGIDs) are common comorbidities in CFS which are characterized by diarrhea, abdominal pain, nausea and exclusion of organic gastrointestinal disease. It has been reported that more than 35% of CFS patients have FGIDs, which has greater functional impairment [2–4]. Significantly, patients with isolated peptic ulcer and inflammatory bowel disease have a higher risk of CFS than patients with isolated nonorganic gastrointestinal diseases such as depression and diabetes [5, 6]. It suggests that gastrointestinal dysfunction plays an important role in CFS.

Symptomatic treatment has been considered as the therapeutic principle for the treatment of CFS [7]. CFS with sleep disorder is mainly treated by cognitive behavioral therapy (CBT) which is a recognized treatment for CFS [8]. Cognitive impairment is mainly treated with CBT, modafinil, and caffeine [9]. For musculoskeletal pain, massage, myofascial release, acupuncture, meditation and relaxation are mainly used. However, in the National Institute for Health and Clinical Excellence (NICE) version of the CFS diagnosis and management guide, there are few management measures for gastrointestinal symptoms in CFS [10]. A few studies reported the beneficial effect of gastrointestinal management on relieving fatigue and restoring energy in CFS patients [11, 12]. In the context of dietary and nutrition interventions, multi mineral and vitamin supplements seem to help manage CFS fatigue, neurocognitive symptoms, depression and sleep disorders [13]. In addition to nutritional supplements, using probiotics is considered to be an effective way to treat CFS [14, 15]. But these studies pay less attention to whether the therapies are beneficial in gastrointestinal function with CFS patient,

and whether the relieving effect on fatigue of CFS patients is related to the improvement of gastrointestinal function. Therefore, it is necessary to further clarify the effects and approaches of gastrointestinal management related therapies on CFS, in order to treat patients with CFS better.

Prolong-life-with-nine-turn-method (PLWNT) is a physical and mental exercise in China. It is an effective method to regulate gastrointestinal symptoms by regulating the mind, breathing and cooperating with eight self-rubbing methods and one upper body rocking. Siwei found that PLWNT can further improve the symptoms of functional dyspepsia on the basis of drugs [16]. The effect of PLWNT on functional dyspepsia is related to further reducing plasma somatostatin content and increasing motilin content, which will promote gastrointestinal peristalsis to improve related symptoms [17, 18]. Our team early research found that 12-week PLWNT exercise can effectively improve fatigue, sleep quality, anxiety and depression of CFS patients [19]. However, it is lack of research to explore the therapeutic effect of PLWNT on gastrointestinal symptoms in CFS and the therapeutic effect of PLWNT on CFS needs to be further verified. Moreover, it is unclear that whether the therapeutic effect of PLWNT is related to the improvement of gastrointestinal function in CFS.

In view of the close relationship between gastrointestinal function and the CFS, we hypothesized that PLWNT is not inferior to CBT in the treatment of CFS, and has more advantages in improving gastrointestinal symptoms of CFS. This advantage may be an important way for PLWNT to treat CFS. In this study, PLWNT and CBT is used to treat CFS patients. During the treatment, self-report scales and intestinal flora detection are used to evaluate the fatigue and gastrointestinal function to clarify the specific role of PLWNT. The study will explore the potential association between fatigue, gastrointestinal function and treatment process in CFS. It will provide effective treatment measures for CFS patients, especially those with gastrointestinal symptoms. And more patients with CFS associated with gastrointestinal symptoms will benefit from this.

## Methods

### Design

The study is a non-inferiority randomized controlled trial (RCT). A total of 96 CFS patients will be recruited and randomly divided into the intervention group and the control group at a ratio of 1:1. Participants of the intervention group will be treated with PLWNT and participants of the control group will be treated with CBT. The whole study period will be 38-week including baseline, 12-week treatment and 6 months follow-up period. The study design is illustrated in the flow chart in (Fig 1).

### Participant

Patients will be recruited in Yueyang Hospital of Integrated Traditional Chinese and Western Medicine, through bulletin board posts and in Shanghai University of traditional Chinese medicine and surrounding communities through advertising (posting notices and online advertisement). All eligible participants will be tested in Yueyang Hospital and operated by doctor responsible for recruitment.

**Inclusion criteria.** (1) Middle aged and young people aged 20 to 60; no gender restrictions

(2) Participants accord with the Fukuda criteria [20], which require at least six months of fatigue with unexplained by clinical evaluation, together with the concurrent at least four of the following eight items: (a) The severity of memory or attention decline leads to a substantial decline in professional ability, educational ability, social activity ability and personal life ability

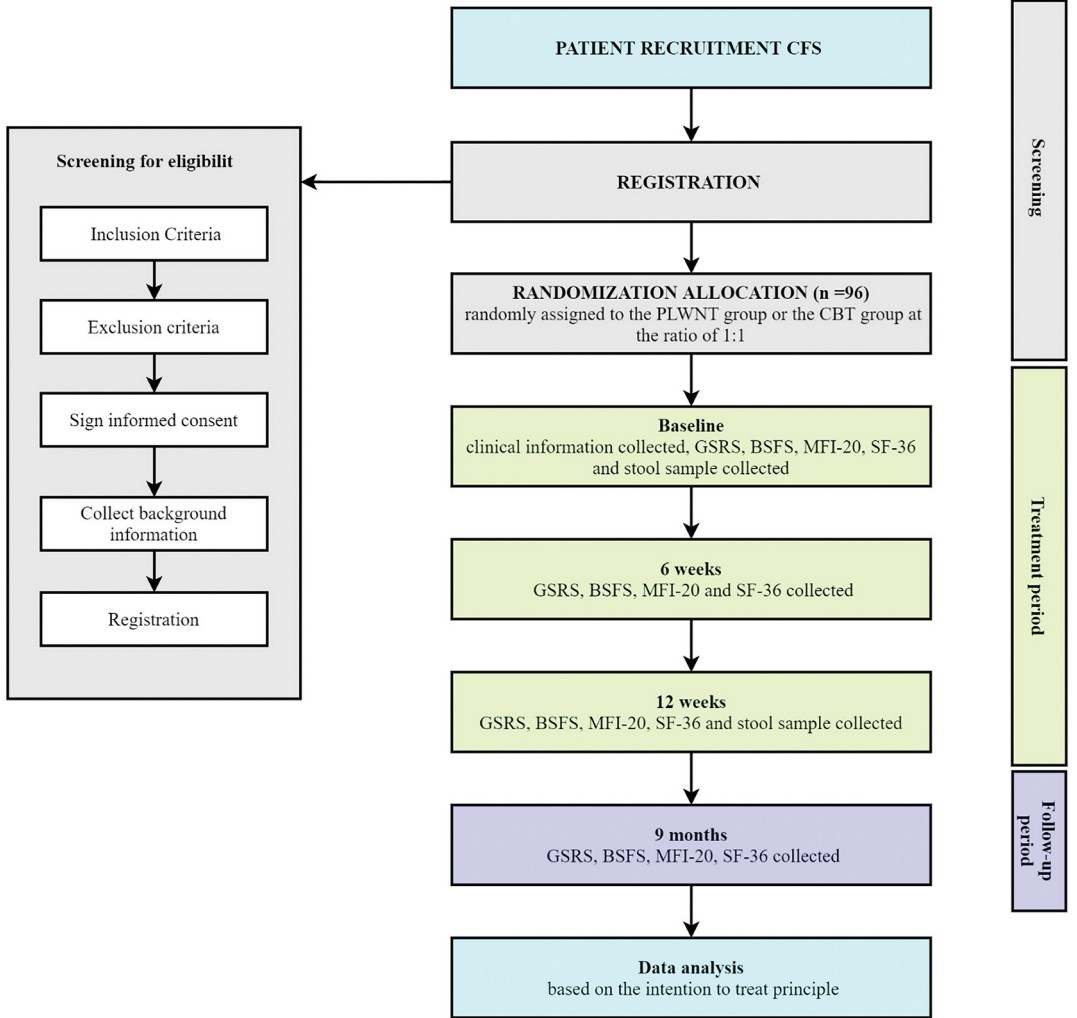

**Fig 1. Study flow chart.**

compared with that before the disease (b) Sore throat(c) Neck or axillary lymph node tenderness(d) Muscle pain (e) No redness, swelling, multiple joint pain(f) Headache with different attack mode, type and severity (g) Unable to recover energy after sleep (h) Myalgia after fatigue for more than 24 hours

(3) Be willing to participate and sign the informed consent form after knowing the whole research scheme.

**Exclusion criteria.**  (1) Patients have taken drugs (especially antibiotics, steroid hormone) that may affect the judgment of results within one month before the study

(2) Definite diagnosis of gastrointestinal organic diseases, liver and kidney dysfunction, tumors or serious cardiovascular and cerebrovascular diseases, endocrine system diseases, sports system diseases, autoimmune diseases, infectious diseases, diabetes or other mental diseases.

(3) Patients with significant dietary changes before this study

(4) Patients who have received radiotherapy, chemotherapy or surgery

(5) Patients with drug addiction, heavy metal poisoning and other similar situations

(6) Pregnant and lactating women

## Intervention

The intervention schedules for each group are shown in Fig 2 Intervention Schedules

**CBT intervention.** *CBT procedure.* CBT training includes a half-day seminar hosted by experts in related fields and 12 weeks of CBT courses under the guidance of experts.

Before treatment, participants will receive relevant knowledge about CFS and CBT. This will help CFS patients understand the principle and role of cognitive therapy in the treatment of CFS.

Within the formal 12-week intervention, psychologist will be invited to give lectures to participants of CBT on the prevention and treatment of CFS and psychological counseling once a week for one hour. For the remaining six days, psychological experts will distribute relevant learning materials and assign homework according to the situation of the subjects.

*CBT measures.* CBT intervention will be conducted by a psychotherapist who has been engaged in psychotherapy for more than 5 years. CFS patients will be treated according to the psychotherapy Manual of CFS [21, 22]. Interventions include: (a) understanding CFS based on physiological and psychological knowledge; (b) agreement on treatment goals; (c) response of CFS; (d) avoidance of excessive activity or rest; (e) planned exposure; (f) Standard CBT procedure [23].

**PLWNT intervention.** *PLWNT procedure.* PLWNT training includes a half-day seminar conducted by a teacher from Shanghai University of Traditional Chinese Medicine who has been teaching Qigong for more than 5 years. The 12-week PLWNT practice will be conducted under the guidance of a teacher.

At initial seminars, participants will receive relevant training about PLWNT, including the introduction of the origin, characteristics and relevant precautions of PLWNT, information of its function and action process in order to help them understand and practice the PLWNT. Participants will be asked to perform abdominal breathing and pay attention to the umbilicus with consciousness, to achieve the role of "regulating body, breath, mind". This process will continue until they could cooperate with other movements. This process will be taught by the coach to ensure the correct movement.

| Period | Screening | Allocation | Intervention | | | Follow up |
|---|---|---|---|---|---|---|
| Week | -2 week | -1 week | 0 week | 6 week | 12 week | |
| Eligibility screening | √ | | | | | |
| Informed consent | √ | | | | | |
| Demographics | √ | | | | | |
| Diagnosis | √ | | | | | |
| Medical history | √ | | | | | |
| Randomization and allocation | | √ | | | | |
| MFI-20 | | | √ | √ | √ | √ |
| SF-36 | | | √ | √ | √ | √ |
| GSRS | | | √ | √ | √ | √ |
| BSFS | | | √ | √ | √ | √ |
| Intestinal flora detection | | | √ | | √ | |
| Adverse events | | | | √ | √ | √ |

**Fig 2. Intervention schedules.**

After the seminars, the coach will teach for one hour at a fixed time every week. Each course includes 10 minutes of warm-up, skill training for 40 minutes and a cooling down portion for 10 minutes. The coach will conduct 12-week intervention at Shanghai University of Traditional Chinese Medicine, once a week, one hour at a time. In addition, researcher will supervise the participants to practice at home for 30 minutes for the remaining six days of the week on WeChat.

*PLWNT measures.* PLWNT is performed according to the ancient Chinese document *Yi Shen Ji* (Fig 3) [24].

Step one: The middle three fingers of two hands press Jiuwei (CV15) and rub it clockwise for 21 times, within3 min.

Step two: The middle three fingers of two hands rub Jiuwei(CV15) clockwise and go down to Qugu (CV2) for 21 times, within3 min.

Step three: The middle three fingers of two hands rub Qugu(CV2) clockwise and go up to Jiuwei (CV15) for 21 times, within3 min.

Step four: The middle three fingers of two hands directly push from Jiuwei (CV15) to Qugu (CV2) for 21 times, within3 min.

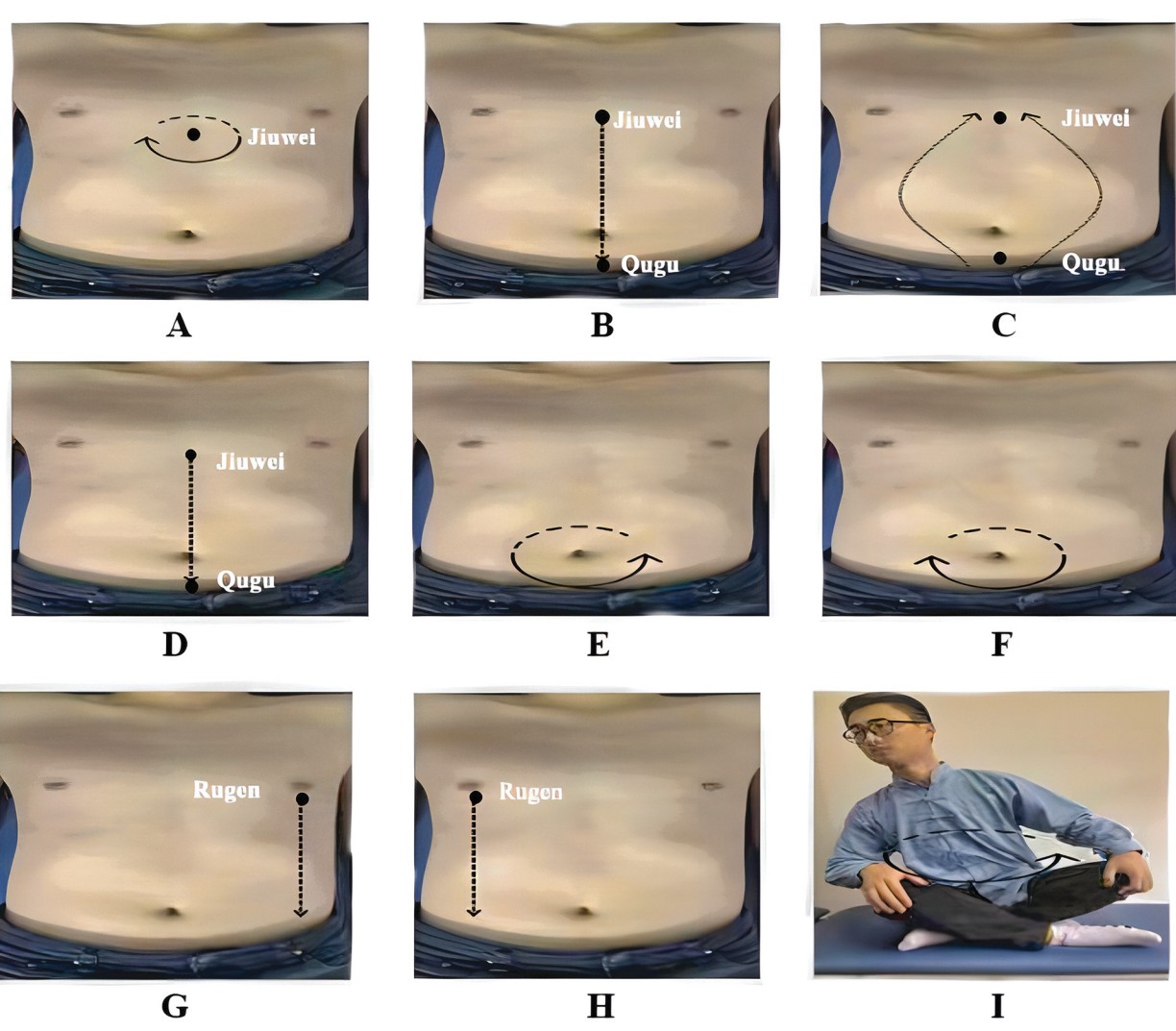

**Fig 3. PLWNT operation diagram.**

Step five: The right hand rubs the abdomen around the navel clockwise for 21 times, within3 min.

Step six: The lefthand rubs the abdomen around the navel counterclockwise for 21 times, within3 min.

Step seven: Left hand with the thumb forward and the other four fingers backward hold on left renal region and the middle three fingers of right hand directly push from Rugeng (ST14) to groin for 21 times, within3 min.

Step eight: Right hand with the thumb forward and the other four fingers backward hold on right renal region and the middle three fingers of left hand directly push from Rugeng (ST14) to groin for 21 times, within3 min.

Step nine: sit on the ground with the legs crosses and the hands on the knees. Rotate the upper body clockwise 21 times and then counterclockwise 21 times.

The acupoints involved in Qigong are on Ren channel, Spleen Meridian and Stomach Meridian. According to WHO standards, Jiuwei (CV15) is on upper abdomen, anterior mid-line and 1 cun below the xiphostern. Qugu (CV2) is on lower abdomen, anterior midline and upper margin of pubic bone. Rugeng (ST14) is on the chest, the fifth rib space and 4 cun beside the anterior midline [25].

**Concomitant interventions.** All other treatments for CFS will be banned during the study, including drug of non-drug. They may receive treatment which is not related to CFS. Any change of concurrent treatments will be recorded.

## Outcomes measures

The outcomes measures include related self-assessment scales, such as the Multidimensional Fatigue Inventory-20 (MFI-20), the Gastrointestinal Symptom Rating Scale (GSRS), the Short Form 36-item Health Survey (SF-36), and Bristol Stool Form Scale (BSFS). The assessments will be carried out at week 0 (baseline), week 6 (midterm), week 12 (endpoint), and week 38 (follow-up). The primary outcome measure is the score of GSRS after 12 weeks of intervention. To explore the effect of PLWNT, the related indexes of gastrointestinal is also studied. Fecal samples are collected to analyze the diversity of intestinal flora. All measurements and measuring time points are shown in Fig 2.

**General information collection.** General information data include age, gender, body mass index, stature, education degree, occupation, working hours, and sleep time.

**Primary outcomes.** *MFI-20.* MFI-20 is the preferred tool for evaluating CFS in recent years [26].MFI-20 will be used to assess the degree of fatigue, which includes 20 items in five dimensions: integrity, physiology, spirit, activity and enthusiasm. Likert 5-level scoring method is adopted for each item, with 20 points for complete compliance and 6 points for complete non-compliance. Among them, 10 items describing fatigue are scored positively and 10 items describing non fatigue are scored reversely, with a total score of 20–100. The higher the score, the heavier the fatigue degree [27].

**Secondary indicators.** *GSRS.* GSRS is a scale to evaluate the severity of gastrointestinal symptoms of digestive diseases especially FGIDs including irritable bowel syndrome, func-tional dyspepsia and other digestive diseases [28, 29]. Cronbach's α is 0.896, Guttman's half-factor is 0.920, and the content validity index of each item is 0.78~1.00 in Chinese version of GSRS [30]. GSRS includes 15 items in five symptom clusters: reflux, abdominal pain, indiges-tion, diarrhoea and constipation. Likert 7-level scoring method is adopted for each item, with 1 point for no gastrointestinal symptoms at all and 7 points for the severest gastrointestinal symptoms. The severer the symptoms, the higher the score [31].

*BSFS*. BSFS is the recommended tool for evaluating the consistency of feces [32], which categorizes stools into seven types ranging from type 1 (hard lumps) to type 7 (watery diarrhea).

*SF-36*. SF-36 is a concise health survey questionnaire developed by Boston Institute of health in the United States. SF-36is the most common scale to measure the general health status of CFS patients, which can effectively distinguish the CFS population from the healthy population [26]. There are 36 items in total and 9 dimensions including physiological function, physical pain, general health, vitality, social function, emotional function, mental health and health change. Each item of the nine dimensions is coded and summed up separately and expressed by 0–100. The higher the score, the better the possible health status [26].

*Fecal sampling*. Fresh stool samples will be collected from participants in the hospital and frozen in a refrigerator at -80°C within 3 hours. The E.Z.N.A. soil DNA kit (Omega Bio-Tek, Norcross, GA, USA) will be used for microbial DNA extraction. The V3-V4 hypervariable regions of 16S rRNA gene of intestinal flora will be amplified by PCR. Primers will be used 338F (5′–ACTCCTACGGGAGGCAGCAG–3′) and 806R (5′–GGACTACHVGGGTWTCTAAT–3′). Illumina MiSeq PE300 will be used for sequencing. Raw data will be uploaded to NCBI SRA database.

## Participant safety

When patients participate in this study, they will be required to know the scheme of the whole study and sign the informed consent. To prevent and better treat any injury that may be caused by this study, the researcher will detect any potential adverse events (AEs) and truthfully record on the CRF form. In this study, AEs are mainly related to PLWNT, including muscle soreness, dizziness, tinnitus, headache, shortness of breath, palpitations, irritability and hallucinations. If an adverse event occurs, it will be dealt with according to the emergency plan. Researchers will initially determine the severity of AEs. Minor AEs will be treated by the attending physician. Serious AEs will be reported by the researcher to the ethics committee. The correlation between the event and intervention and severity will be evaluated.

## Quality control

The head of the research center will be responsible for the design, coordination and quality control in whole study. All researchers will receive uniform training before the period of data collection. Before the data collection period, patients were required to maintain a healthy lifestyle, such as avoiding late nights, drinking alcohol and skipping breakfast so as to prevent patients from being affected by unstable factors. Relevant data collection will be completed between 7:00–9:00 in the morning. Throughout the research period, a data monitoring of committee independent of researchers and sponsors will extract 10% of case reports from and check the data every three months. The committee will check logic problems, determination of test values, abnormal safety indicators after treatment, vacancy values, compliance, standardization, integrity, consistency, etc. The committee will consist of clinicians, statistician, microbiologist, psychologist, and ethicist.

## Follow up

To evaluate the long-term efficacy and safety of CBT and PLWNT, the research will follow up the patients after 6 months of the treatment. Participants will not receive any intervene during follow-up. At the end of follow-up, the investigator will make a contract with patients by telephone, E-mail and etc. They will be asked to come to reception room and fill in the GSRS, MFI-20, SF-36 scale.

## Sample size

In this study, the sample size is calculated based on the improvement of fatigue scale score, hypothesis:

$$H_0 : \mu_1 - \mu_2 \leq \Delta$$

$$H_1 : \mu_1 - \mu_2 \leq \Delta$$

$$n = 2 \times \frac{\left(Z_{\alpha/_2} + Z_\beta\right)^2 \times^2}{(\delta + \Delta)^2}$$

$\mu_1$ is the average score of fatigue scale after 12 weeks of intervention in the PLWNT group, and $\mu_2$ is the average score of fatigue scale after 12 weeks of intervention in the CBT group. According to the preliminary research results [33], the standard deviation σ was 3.172, the variation difference δ was 2.216 and non-inferiority margin Δ was 0. According to the formula, among α = 0.05 (both sides) and 1 − β = 0.90. The sample size of each group is calculated to be 43. Considering a 10% dropout rate, the final sample size was set to 48 each group.

## Randomization and allocation concealment

The patients who meet the diagnosis of CFS and signed informed consent will be randomly assigned to control group and intervention group at a ratio of 1:1. Randomization will be computer generated by independent and uninformed statistical experts. Randomization sequence will be created using SPSS Software (SPSS 24.0, SPSS Inc, Chicago, IL, USA) and will be stratified by center with severity of gastrointestinal symptom, severity of fatigue and age. Managers who do not participate in the recruitment and treatment place each number in an opaque envelope and keep it. After screening the patients, the doctor responsible for recruitment will contact the administrator who will send the file to the eligible participants and inform them of the allocation.

## Blinding

Due to the nature of the intervention, CBT and PLWNT therapists and participants cannot be blinded. During the study, statisticians, administrator, data collectors and result evaluator will be blinded. After the statistics, the blinding method will be broken.

## Data collection and management

Two data administrators who do not belong to the research team and blinded to group allocation will be responsible for data entry and database establishment. All original data related to the study will be stored at Shanghai University of Traditional Chinese Medicine and uploaded in real time to the China Clinical Trial Registry.

## Statistical analysis

**Statistical analysis of scales and general information.** The data analysis will be based on the intention to treat principle and per-protocol analysis. Regardless of whether they completed the trial or what treatment they received, the participate will be kept in the group after randomization for result analysis. Participants who failed to complete the study will be always considered as unchanged from the last observation.

All statistical analyses will be used for SPSS software by independent and uninformed statisticians. "Mean ± standard deviation" is used for the measurement data of normal distribution, "median and interquartile spacing" is used for the measurement data that do not conform to normal distribution, and "frequency and percentage" are used for the counting data. Inspection standard $\alpha$ equals to 0.05. P less than 0.05 means the difference is statistically significant.

Student's t test or Mann–Whitney test will be used for continuous variables to compare the characteristics of the two groups at baseline. Categorical variables will be analyzed by chi square test or Fisher's exact test to compare the baseline situation of the two groups. The efficacy will be measured at four time points. Evaluation using repeated measurement analysis of variance or generalized estimation equations.

**Statistical analysis of intestinal flora.**   In intestinal flora analysis, the cluster analysis of operational taxonomic units (OTUs) will be carried out. Alpha and beta diversity of intestinal microorganisms were calculated using Mothur software package [34]. PCoA analysis based on bray-curtis distance algorithm (principal coordinate analysis) will be used to test the similarity of microbial community structure among samples. Nonparametric test will be used to analyze whether the differences in microbial community structure between sample groups is significant. Linear discriminant analysis Effect Size(LEfSe) (LDA>2, P<0.05) will be used to identify bacterial groups with significant differences in abundance from the phyla to genus level among different groups.

### Ethics statement

The study protocol was approved by the Regional Ethics Review Committee of Yueyang Hospital (number: 2022–007). All participants will provide their written consent. Trial registration: The trial was registered at Chinese Clinical Trial Registry http://www.chictr.org.cn/showproj. aspx?proj=151456

(Registration No.: ChiCTR2200056530). Date: 2022-02-07.

### Discussion

To the best of our knowledge, this is the first study to explore the effect of Qigong on CFS from the perspective of gastrointestinal function. The purpose of study is explore the effect of PLWNT on the fatigue and gastrointestinal of CFS, which will define the specific role of PLWNT in treatment of CFS. Self-report scales will help to comprehensively evaluate the gastrointestinal function and fatigue level of CFS. The intestinal flora will be detected by 16SrDNA technology which is the most suitable method for identifying bacterial classification and clarifying the correlation among microorganisms, hosts, diseases.

Gastrointestinal function is closely related to the level of fatigue in CFS patients. Relevant research shows that the patient with CFS has intestinal barrier dysfunction, gut immune dysfunction, altered gut motility, altered gastrointestinal secretion, altered intestinal flora, and altered gut-brain axis [35, 36].These changes make CFS patients prone to diarrhea, abdominal pain, constipation and other digestive symptoms [37–41]. At the same time, some studies found that abnormal alteration to intestinal flora are linearly related to decreased motivation, increased levels of fatigue, anxiety, depression in CFS patient [42, 43]. Some scholars believe that this is related to intestinal inflammation, intestinal barrier disorder, and changes in gut brain axis caused by intestinal flora imbalance [35, 44, 45]. These changes in gastrointestinal function are the pivotal participators in the CFS pathogenesis, which aggravates the fatigue of CFS patients by activating the corresponding brain areas and increasing the level of inflammation in the body confirmed by cohort study and clinical study [15, 46].

PLWNT is a traditional Chinese medicine mind-body exercise which put emphasis on regulating gastrointestinal function. The practice of PLWNT include "regulating mind, breath and body". In PLWNT, regulating the body include eight kinds of abdominal massage. The action is operated at abdominal acupoints including Jiuwei (CV15), Qugu (CV2), Rugeng (ST14) by pushing, holding and rubbing. Systematic review shown that abdominal massage can enhance gastrointestinal peristalsis, alleviate gastrointestinal symptoms and improve digestive system symptoms, such as diarrhea, nausea and vomiting [47, 48]. The mechanism is to stimulate parasympathetic excitation by increasing the secretion of substances such as gastric acid and gastrin [49]. In terms of respiratory regulation, participant will be required to adopt abdominal breathing which is defined as breathing in slowly and deeply through consciously increasing diaphragm activity. This type of breathing will increase intra-abdominal pressure and stimulate the phrenic and vagus nerves that control the movement of the diaphragm, which affects the movement of the colon and activates parasympathetic nerves in the brain [50]. The mechanism of regulating mind in PLWNT is similar to CBT. CBT is a well-established treatment for CFS that develops a person's ability to focus on the present moment rather than acting on "automatic thinking" [23, 51]. It has been reported that CBT, as a brain-targeted intervention, can improve persistent gastrointestinal symptoms, which may be related to the top-down signals in the brain regulating changes in intestinal flora [52, 53].

In conclusion, PLWNT is an effective method to regulate gastrointestinal function. Its therapeutic effect on CFS is probably related to regulating gastrointestinal function. This study will provide evidence the effect of PLWNT on fatigue and gastrointestinal function in patients with CFS and explore the intervention pathway of PLWNT.

## Limitation

Although this is a promising study, it also has some limitations: In this study, the intervention is a non-drug therapy, which cannot be ideally implemented in accordance with the provisions of double-blind. However, the strict management of researchers, data managers and statisticians in this study will help us to reduce performance bias. Secondly, there are great differences in gastrointestinal function among different CFS patients, because CFS is a physical and mental disease with multisystem symptom. To increase the CFS patient homogeneity, we assess baseline weight and BMI in CFS patients. Finally, the treatment cycle of PLWNT for CFS is still in the exploration stage. Therefore, according to previous PLWNT studies on functional dyspepsia [16] and CFS related research [54, 55], the study will evaluate at 0-week, 6 weeks and the 12 weeks during the treatment and six months after the treatment as a follow-up. It will further clarify the short-term and long-term effects of PLWNT and understand the therapeutic effect of PLWNT.

## Supporting information

**S1 File. Reporting checklist for protocol of a clinical trial.**
(DOCX)

**S2 File. Ethics statement document.**
(DOCX)

**S3 File. Ethical approval of clinical research plan.**
(DOCX)

**S4 File. Study protocol proofs.**
(PDF)

## Acknowledgments

The authors appreciate their colleagues in the YueYang Hospital of Integrated Traditional Chinese and Western Medicine, Shanghai University of Traditional Chinese Medicine, Shanghai Municipal Hospital of Traditional Chinese Medicine, for their full support in recruiting and treating patients. The authors appreciate the efforts and help of investigators participating in this trial. The authors appreciate Wei Deng for her guidance.

## Author Contributions

**Conceptualization:** Yanli You, Fei Yao.

**Data curation:** Chaoqun Xie.

**Investigation:** Guangxin Guo, Fangfang Xie, Chong Guan, Yanbin Cheng.

**Methodology:** Qing Ji.

**Project administration:** Fei Yao.

**Writing – original draft:** Yuanjia Gu.

**Writing – review & editing:** Yanli You, Fei Yao.

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
