## [Decision Letter · Decision Letter 0]

6 Mar 2023

PONE-D-22-28257Effect of Prolong-life-with-nine-turn-method (Yan Nian Jiu Zhuan) Qigong on Fatigue and Gastrointestinal function in Patients with Chronic Fatigue Syndrome: Study Protocol for a Randomized Controlled TrialPLOS ONE

Dear Dr. Yao,

Thank you for submitting your manuscript to PLOS ONE. After careful consideration, we feel that it has merit but does not fully meet PLOS ONE’s publication criteria as it currently stands. Therefore, we invite you to submit a revised version of the manuscript that addresses the points raised during the review process.

We look forward to receiving your revised manuscript.

Kind regards,

Walid Kamal Abdelbasset, Ph.D.

Academic Editor

PLOS ONE

Journal Requirements:

2. "PLOS requires an ORCID iD for the corresponding author in Editorial Manager on papers submitted after December 6th, 2016. Please ensure that you have an ORCID iD and that it is validated in Editorial Manager. To do this, go to ‘Update my Information’ (in the upper left-hand corner of the main menu), and click on the Fetch/Validate link next to the ORCID field. This will take you to the ORCID site and allow you to create a new iD or authenticate a pre-existing iD in Editorial Manager. Please see the following video for instructions on linking an ORCID iD to your Editorial Manager account: " ext-link-type="uri" xlink:type="simple">https://www.youtube.com/watch?v=_xcclfuvtxQ"

3. We note that the original protocol file you uploaded contains a confidentiality notice indicating that the protocol may not be shared publicly or be published. Please note, however, that the PLOS Editorial Policy requires that the original protocol be published alongside your manuscript in the event of acceptance. Please note that should your paper be accepted, all content including the protocol will be published under the Creative Commons Attribution (CC BY) 4.0 license, which means that it will be freely available online, and any third party is permitted to access, download, copy, distribute, and use these materials in any way, even commercially, with proper attribution.

Therefore, we ask that you please seek permission from the study sponsor or body imposing the restriction on sharing this document to publish this protocol under CC BY 4.0 if your work is accepted. We kindly ask that you upload a formal statement signed by an institutional representative clarifying whether you will be able to comply with this policy. Additionally, please upload a clean copy of the protocol with the confidentiality notice (and any copyrighted institutional logos or signatures) removed.

Reviewers' comments:

Reviewer's Responses to Questions

**Comments to the Author**

1. Does the manuscript provide a valid rationale for the proposed study, with clearly identified and justified research questions?

Reviewer #1: Partly

Reviewer #2: Partly

Reviewer #3: Yes

2. Is the protocol technically sound and planned in a manner that will lead to a meaningful outcome and allow testing the stated hypotheses?

Reviewer #1: Partly

Reviewer #2: Yes

Reviewer #3: Yes

3. Is the methodology feasible and described in sufficient detail to allow the work to be replicable?

Reviewer #1: No

Reviewer #2: No

Reviewer #3: Yes

4. Have the authors described where all data underlying the findings will be made available when the study is complete?

Reviewer #1: No

Reviewer #2: Yes

Reviewer #3: No

5. Is the manuscript presented in an intelligible fashion and written in standard English?

Reviewer #1: Yes

Reviewer #2: Yes

Reviewer #3: Yes

6. Review Comments to the Author

You may also provide optional suggestions and comments to authors that they might find helpful in planning their study.

Reviewer #1: The authors have stated that this is a non-inferiority study, however the power calculations haven't accounted for the inferiority margin. Which suggests it's a superiority study.

The non-inferiority margin needs to be defined and update hypotheses which currently suggest superiority.

Explicitly define the population, it says ITT. I.e analysed according to randomised group e

Regardless of what treatment they received.

State results will be reported according to CONSORT guidelines.

Simple randomisation has been used. Did the authors not consider one or two stratification factors known to be prognostic of CFS, e.g age or severity of comorbidities.

Reviewer #2: research should be replicable. but i cant understand the the prolong life with nine turn method, you should describe it in all details, this article could help you it was the same

Xie F, You Y, Guan C, Xu J, Yao F. The Qigong of Prolong Life With Nine Turn Method Relieve Fatigue, Sleep, Anxiety and Depression in Patients With Chronic Fatigue Syndrome: A Randomized Controlled Clinical Study. Front Med (Lausanne). 2022 Jun 30;9:828414. doi: 10.3389/fmed.2022.828414. PMID: 35847786; PMCID: PMC9280429.

Reviewer #3: Keywords: use MeSH keywords

Abstract:

1. Include brief introduction about CFD and PLWNT procedure.

2. Mention the study design, study duration, and study setting.

3. Mention the randomization and allocation details.

4. Mention the duration of measuring the outcome variables.

5. Mention the statistical tests performed for the study.

Manuscript

6. How come this study is differing from the reference number 16 - 19?

7. The novelty of the study is missing, including more recent references emphasizing the need for this study.

8. Mention the gaps monitored by the researcher in the previous studies.

9. Include the clinical significance of this study over clinicians, patients, and researchers after the study hypothesis.

10. Include the abbreviations of acronyms when it is used for the first time.

11. Mention who included the study participants in the trial?

12. Mention about the informed consent information.

13. Mention in detail about the randomization and allocation procedure.

14. Include the intervention program PLWNT with intensity, frequency, mode and duration.

15. Include the outcome measures and its reliability and validity.

16. The statistical tests used for the study was not apt to this study.

I look forward to reading the revised version of the manuscript.

Thanks again, and good luck with researching in this challenging time.

7. PLOS authors have the option to publish the peer review history of their article (what does this mean?). If published, this will include your full peer review and any attached files.

Reviewer #1: No

Reviewer #2: **Yes: **Ibrahim Dewir

Reviewer #3: **Yes: **Gopal Nambi

---

## [Author Response · Author response to Decision Letter 0]

24 Mar 2023

Reviewer #1

Response

Q: (1) The authors have stated that this is a non-inferiority study, however the power calculations haven't accounted for the inferiority margin. Which suggests it's a superiority study. The non-inferiority margin needs to be defined and update hypotheses which currently suggest superiority.

A: Thank you very much for your suggestion. According to your suggestion, we have modified the formula for calculating the sample size to conform to the design of non-inferiority. Modify to n=2×((Z_(α⁄2)＋Z_β )^2×�^2)/(（〖δ+Δ）〗^2 ). the standard deviation � was 3.172, the variation difference δ was 2.216 and non-inferiority margin Δ was 0. After modification, the sample size remains the same as the previous sample size. The final sample size was set to 48 each group. At the same time, we further define the hypotheses. This study hypothesizes that PLWNT is not inferior to CBT in the treatment of CFS and has more advantages in improving gastrointestinal symptoms of CFS. This advantage may be an important way for PLWNT to treat CFS.

Q: (2) Explicitly define the population, it says ITT.

A: Thank you very much for your suggestion, which makes our article more rigorous and complete. As supplied by our article on page13, the population definition of ITT has been marked in red. The specific description is as follows: 

The data analysis will be based on the intention to treat principle and per-protocol analysis. Regardless of whether they completed the trial or what treatment they received, the participate will be kept in the group after randomization for result analysis. Participants who failed to complete the study will be always considered as unchanged from the last observation. State results will be reported according to CONSORT guidelines. 

Q: (3) Simple randomization has been used. Did the authors not consider one or two stratification factors known to be prognostic of CFS, e.g age or severity of comorbidities.

A: Thank you for your valuable comments, which make our research more complete. As described by our article on page12, the supplement of randomization has been marked in red. The specific description is as follows:

The patients who meet the diagnosis of CFS and signed informed consent will be randomly assigned to control group and intervention group at a ratio of 1:1. Randomization will be computer generated by independent and uninformed statistical experts. Randomization sequence will be created using SPSS Software (SPSS 24.0, SPSS Inc, Chicago, IL, USA) and will be stratified by center with severity of gastrointestinal symptom, severity of fatigue and age. Managers who do not participate in the recruitment and treatment place each number in an opaque envelope and keep it. After screening the patients, the doctor responsible for recruitment will contact the administrator who will send the file to the eligible participants and inform them of the allocation.

Reviewer #2

Response

Q: (1) research should be replicable. but i cant understand the the prolong life with nine turn method, you should describe it in all details, this article could help you it was the same.

A: We are very honored by your interest in the treatment of PLWNT. The article “The Qigong of Prolong Life With Nine Turn Method Relieve Fatigue, Sleep, Anxiety and Depression in Patients With Chronic Fatigue Syndrome: A Randomized Controlled Clinical Study” is a preliminary study by our team. The operation of PLWNT in this paper is consistent with the previous research. In order to further standardize the operation position and action, this paper uses acupoints to describe. At the same time, we uploaded the relevant operation flow diagram (Fig.3) to help understand (the uploaded picture is consistent with the previous research and has been approved by the previous researchers). Based on previous studies, this study further clarified the effects of PLWNT on gastrointestinal symptoms in patients with CFS. Based on the effective effect of PLWNT on gastrointestinal function, this study explores the key factors of PLWNT in the treatment of CFS from the perspective of gastrointestinal function.

Reviewer #3

Response

Q: (1) Keywords: use MeSH keywords

A：Thank you for your valuable comments, which make our research more complete. According to your suggestion, we modified the key words of the article. The modified part is marked in red: 

Keywords: Fatigue Syndrome, Chronic; Signs and Symptoms, Digestive; Qigong, Cognitive Behavioral Therapy.

Q: Abstract: 

（1）Include brief introduction about CFD and PLWNT procedure.

（2）Mention the study design, study duration, and study setting.

（3）Mention the randomization and allocation details.

（4）Mention the duration of measuring the outcome variables.

（5）Mention the statistical tests performed for the study.

A: Thank you for your valuable comments. Through your suggestion, we have carefully read the full text of the manuscript and found that some descriptions are indeed incomplete and accurate. We have modified the article summary according to your suggestions, and the modified places are indicated in red. The details are as follows:

Introduction: Chronic fatigue syndrome (CFS) is a physical and mental disorder in which long-term fatigue is the main symptom. CFS patients are often accompanied by functional gastrointestinal diseases (FGIDs), which lead to decreased quality of life and increased fatigue. Prolong-life-with-nine-turn-method (PLWNT) is a kind of physical and mental exercise. Its operation includes adjusting the mind, breathing and cooperating with eight self-rubbing methods and one upper body rocking method. PLWNT was used to improve the digestive function in ancient China and to treat FGIDs such as functional dyspepsia and irritable bowel syndrome in modern times. Previous studies have shown that PLWNT can reduce fatigue in patients with CFS. But it is unclear whether the effect of PLWNT on CFS fatigue is related to gastrointestinal function. The aim of this study was to explore the relationship between PLWNT and fatigue and gastrointestinal function in patients with CFS.

Methods: This study is a non-inferiority randomized controlled trial (RCT). The whole study period is 38 weeks, including 2 weeks of baseline evaluation, 12 weeks of intervention and 6 months of follow-up. Ninety-six CFS patients will be stratified random assigned to the intervention group (PLWNT) and the control group (cognitive behavior treatment) in the ratio of 1:1 through the random number table generated by SPSS. In the evaluation of results, Multidimensional Fatigue Inventory-20 (MFI-20), Gastrointestinal Symptom Rating Scale (GSRS), Bristol Stool Form Scale (BSFS), and Short Form 36 item health survey (SF-36) will be evaluated at week 0 (baseline), week 6 (midterm), week 12 (endpoint) and month 9 (follow up). The intestinal flora will be evaluated at week 0 (baseline) and week 12 (endpoint). The data results will be processed by statistical experts. The data analysis will be based on the intention to treat principle and per-protocol analysis. In the efficacy evaluation, repeated measurement analysis of variance will be used for data conforming to normal distribution or approximate normal distribution. The data which do not conform to the analysis of repeated measurement variance will be analyzed by the generalized estimation equation. Linear discriminant analysis will be used to clarify the difference species of intestinal flora. The significance level sets as 5%. The safety of interventions will be evaluated after each treatment session.

Discussion: This trial will provide evidence to PLWNT exerting positive effects on fatigue and gastrointestinal function of CFS. It will further explore whether the improvement of PLWNT on CFS fatigue is related to gastrointestinal function. 

Q: (6) How come this study is differing from the reference number 16 - 19?

A: Thank you very much for your reminder, which makes our article more rigorous. References 16-19 are preliminary studies on PLWNT. The research is based on the previous research results. In the early stage, our team has studied the effectiveness and safety of PLWNT in the treatment of CFS, but the therapeutic pathway of PLWNT in treating CFS is still unclear. Due to the operation and clinical efficacy (References 16-18) of PLWNT, PLWNT is considered to be an effective way to adjust gastrointestinal function. Therefore, this study hypothesizes that PLWNT is not inferior to CBT in the treatment of CFS and has more advantages in improving gastrointestinal symptoms of CFS. This advantage may be an important way for PLWNT to treat CFS. 

Q: (7) The novelty of the study is missing, including more recent references emphasizing the need for this study.

A: Thank you very much for your valuable comments. We have added recent references on the necessity of this study to support our study. The study will clarify the effect of PLWNT on fatigue and gastrointestinal function of CFS. It will explore the potential association between fatigue, gastrointestinal function and treatment process in CFS. It will provide effective treatment measures for CFS patients, especially those with gastrointestinal symptoms. And more patients with CFS associated with gastrointestinal symptoms will benefit from this. 

Q: (8) Mention the gaps monitored by the researcher in the previous studies.

A: Thank you very much for your valuable comments. We believe that the gap detected by the researchers in the previous study answers question 6 to a certain extent. Therefore, we further supplemented the relevant research on page 4-5 and marked it with red. The details are as follows: 

Siwei found that PLWNT can further improve the symptoms of functional dyspepsia on the basis of drugs[16]. The effect of PLWNT on functional dyspepsia is related to further reducing plasma somatostatin content and increasing motilin content, which will promote gastrointestinal peristalsis to improve related symptoms[17, 18]. Our team early research found that 12-week PLWNT exercise can effectively improve fatigue, sleep quality, anxiety and depression of CFS patients[19]. However, it is lack of research to explore the therapeutic effect of PLWNT on gastrointestinal symptoms in CFS and the therapeutic effect of PLWNT on CFS needs to be further verified. Moreover, it is unclear that whether the therapeutic effect of PLWNT is related to the improvement of gastrointestinal function in CFS.

Q: (9) Include the clinical significance of this study over clinicians, patients, and researchers after the study hypothesis.

A：Thank you for your suggestion, it makes more sense for us to understand the full text. We added the details on page 5 and marked it in red. The details are as follows:

The study will explore the potential association between fatigue, gastrointestinal function and treatment process in CFS. It will provide effective treatment measures for CFS patients, especially those with gastrointestinal symptoms. And more patients with CFS associated with gastrointestinal symptoms will benefit from this.

Q: (10) Include the abbreviations of acronyms when it is used for the first time. 

A: Thanks for your suggestion. We are very sorry that our mistakes have troubled you. We have reviewed and revised all relevant abbreviations in this article

Q: (11) Mention who included the study participants in the trial?

A: Thanks for your suggestion. The details have been supplemented in the Participant section on page 6. The supplementary contents are as follows: 

All eligible participants will be tested in Yueyang Hospital and operated by doctor responsible for recruitment.

Q: (12) Mention about the informed consent information.

A: Thanks for your suggestion. The details have been supplemented in the Inclusion Criteria section on page 6. The supplementary contents are as follows: 

Be willing to participate and sign the informed consent form after knowing the whole research scheme

Q: (13) Mention in detail about the randomization and allocation procedure.

A: Thanks for your suggestion. The details have been supplemented in the Randomization and allocation concealment section on page 11 and marked it in red. The supplementary contents are as follows: 

The patients who meet the diagnosis of CFS and signed informed consent will be randomly assigned to control group and intervention group at a ratio of 1:1. Randomization will be computer generated by independent and uninformed statistical experts. Randomization sequence will be created using SPSS Software (SPSS 24.0, SPSS Inc, Chicago, IL, USA) and will be stratified by center with severity of gastrointestinal symptom, severity of fatigue and age. Managers who do not participate in the recruitment and treatment place each number in an opaque envelope and keep it. After screening the patients, the doctor responsible for recruitment will contact the administrator who will send the file to the eligible participants and inform them of the allocation.

Q: (14) Include the intervention program PLWNT with intensity, frequency, mode and duration.

A: Thank you for your suggestion, it makes more sense for us to understand the full text. We added the details on page 8 and marked it in red. At the same time, we further supplemented the operation details of PLWNT and uploaded pictures to help understand. Supplement 1 is the operation diagram of PLWNT. The details are as follows:

After the seminars, the coach will teach for one hour at a fixed time every week. Each course includes 10 minutes of warm-up, skill training for 40 minutes and a cooling down portion for 10 minutes. The coach will conduct 12-week intervention at Shanghai University of Traditional Chinese Medicine, once a week, one hour at a time. In addition, researcher will supervise the participants to practice at home for 30 minutes for the remaining six days of the week on WeChat.

PLWNT measures

PLWNT is performed according to the ancient Chinese document Yi Shen Ji. 

Step one: The middle three fingers of two hands press Jiuwei (CV15) and rub it clockwise for 21 times, within3 min.

Step two: The middle three fingers of two hands rub Jiuwei(CV15) clockwise and go down to Qugu (CV2) for 21 times, within3 min.

Step three: The middle three fingers of two hands rub Qugu(CV2) clockwise and go up to Jiuwei (CV15) for 21 times, within3 min.

Step four: The middle three fingers of two hands directly push from Jiuwei (CV15) to Qugu (CV2) for 21 times, within3 min.

Step five: The right hand rubs the abdomen around the navel clockwise for 21 times, within3 min.

Step six: The lefthand rubs the abdomen around the navel counterclockwise for 21 times, within3 min.

Step seven: Left hand with the thumb forward and the other four fingers backward hold on left renal region and the middle three fingers of right hand directly push from Rugeng（ST14）to groin for 21 times, within3 min.

Step eight: Right hand with the thumb forward and the other four fingers backward hold on right renal region and the middle three fingers of left hand directly push from Rugeng（ST14）to groin for 21 times, within3 min.

Step nine: sit on the ground with the legs crosses and the hands on the knees. Rotate the upper body clockwise 21 times and then counterclockwise 21 times.

Q: (15) Include the outcome measures and its reliability and validity.

A: Thank you for your suggestion. The details have been supplemented in the Outcomes measures section on page 9-10. The supplementary contents are as follows:

MFI-20：MFI-20 is the preferred tool for evaluating CFS in recent years[25]

GSRS is a scale to evaluate the severity of gastrointestinal symptoms of digestive diseases especially FGIDs including irritable bowel syndrome, functional dyspepsia and other digestive diseases[27, 28]. Cronbach's α is 0.896, Guttman's half-factor is 0.920, and the content validity index of each item is 0.78~1.00 in Chinese version of GSRS [29]

BSFS is the recommended tool for evaluating the consistency of feces[31], which categorizes stools into seven types ranging from type 1 (hard lumps) to type 7 (watery diarrhea).

SF-36 is a concise health survey questionnaire developed by Boston Institute of health in the United States. SF-36 is the most common scale to measure the general health status of CFS patients, which can effectively distinguish the CFS population from the healthy population[25].

Q: (16) The statistical tests used for the study was not apt to this study.

A: Thank you very much for your valuable comments, which motivates us to explore further in future research. We revised it according to the opinions of statistical experts. The details have been supplemented in the Outcomes measures section on page 9-10. The supplementary contents are as follows:

The data analysis will be based on the intention to treat principle and per-protocol analysis. Regardless of whether they completed the trial or what treatment they received, the participate will be kept in the group after randomization for result analysis. Participants who failed to complete the study will be always considered as unchanged from the last observation. State results will be reported according to CONSORT guidelines. 

All statistical analyses will be used for SPSS software by independent and uninformed statisticians. "Mean ± standard deviation" is used for the measurement data of normal distribution, "median and interquartile spacing" is used for the measurement data that do not conform to normal distribution, and "frequency and percentage" are used for the counting data. Inspection standard α equals to 0.05. P less than 0.05 means the difference is statistically significant.

Student’s t test or Mann–Whitney test will be used for continuous variables to compare the characteristics of the two groups at baseline. Categorical variables will be analyzed by chi square test or Fisher's exact test to compare the baseline situation of the two groups. The efficacy will be measured at four time points. Evaluation using repeated measurement analysis of variance or generalized estimation equations.

Statistical analysis of intestinal flora

In intestinal flora analysis, the cluster analysis of operational taxonomic units (OTUs) will be carried out. Alpha and beta diversity of intestinal microorganisms were calculated using Mothur software package[33]. PCoA analysis based on bray-curtis distance algorithm (principal coordinate analysis) will be used to test the similarity of microbial community structure among samples. Nonparametric test will be used to analyze whether the differences in microbial community structure between sample groups is significant. Linear discriminant analysis Effect Size(LEfSe) (LDA2, P0.05) will be used to identify bacterial groups with significant differences in abundance from the phyla to genus level among different groups.

---

## [Decision Letter · Decision Letter 1]

4 Jun 2023

Effect of Prolong-life-with-nine-turn-method (Yan Nian Jiu Zhuan) Qigong on Fatigue and Gastrointestinal function in Patients with Chronic Fatigue Syndrome: Study Protocol for a Randomized Controlled Trial

PONE-D-22-28257R1

Dear Dr. Fei,

We’re pleased to inform you that your manuscript has been judged scientifically suitable for publication and will be formally accepted for publication once it meets all outstanding technical requirements.

Kind regards,

Huijuan Cao, Ph.D.

Academic Editor

PLOS ONE

Additional Editor Comments (optional):

Reviewers' comments:

Reviewer's Responses to Questions

**Comments to the Author**

1. Does the manuscript provide a valid rationale for the proposed study, with clearly identified and justified research questions?

Reviewer #1: Yes

Reviewer #2: Yes

Reviewer #3: Yes

2. Is the protocol technically sound and planned in a manner that will lead to a meaningful outcome and allow testing the stated hypotheses?

Reviewer #1: Yes

Reviewer #2: Yes

Reviewer #3: Yes

3. Is the methodology feasible and described in sufficient detail to allow the work to be replicable?

Reviewer #1: Yes

Reviewer #2: Yes

Reviewer #3: Yes

4. Have the authors described where all data underlying the findings will be made available when the study is complete?

Reviewer #1: No

Reviewer #2: Yes

Reviewer #3: Yes

5. Is the manuscript presented in an intelligible fashion and written in standard English?

Reviewer #1: Yes

Reviewer #2: Yes

Reviewer #3: Yes

6. Review Comments to the Author

You may also provide optional suggestions and comments to authors that they might find helpful in planning their study.

Reviewer #1: I have no further comments.

Reviewer #2: thanks for great work by author and thanks for correcting all missing parts and illustrate the methodd in details

Reviewer #3: Dear authors,

I really appreciate you for addressing my comments in a very positive manner.

Now the study protocol is potential enough to publish in its current state.

Regards

7. PLOS authors have the option to publish the peer review history of their article (what does this mean?). If published, this will include your full peer review and any attached files.

Reviewer #1: No

Reviewer #2: No

Reviewer #3: **Yes: **Dr. Gopal Nambi, PT, PhD

---

## [Editor Report · Acceptance letter]

28 Jun 2023

PONE-D-22-28257R1 

Effect of Prolong-life-with-nine-turn-method (Yan Nian Jiu Zhuan) Qigong on Fatigue and Gastrointestinal function in Patients with Chronic Fatigue Syndrome: Study Protocol for a Randomized Controlled Trial 

Dear Dr. Yao:

I'm pleased to inform you that your manuscript has been deemed suitable for publication in PLOS ONE. Congratulations! Your manuscript is now with our production department. 

Kind regards, 

on behalf of

Dr. Huijuan Cao 

Academic Editor

PLOS ONE